

# Improved temporal IoT device identification using robust statistical features

Nik Aqil[1], Faiz Zaki[1], Firdaus Afifi[1,2], Hazim Hanif[3], Miss Laiha Mat Kiah[1] and Nor Badrul Anuar[1]

[1] Department of Computer System and Technology, Faculty of Computer Science and Information Technology, Universiti Malaya, Kuala Lumpur, Malaysia

[2] Faculty of Ocean Engineering Technology and Informatics, Universiti Malaysia Terengganu, Kuala Nerus, Terengganu, Malaysia

[3] Department of Software Engineering, Faculty of Computer Science and Information Technology, Universiti Malaya, Kuala Lumpur, Malaysia

Corresponding authors
Nik Aqil, 17072128@siswa.um.edu.my
Faiz Zaki, faizzaki@um.edu.my

## ABSTRACT

The Internet of Things (IoT) is becoming more prevalent in our daily lives. A recent industry report projected the global IoT market to be worth more than USD 4 trillion by 2032. To cope with the ever-increasing IoT devices in use, identifying and securing IoT devices has become highly crucial for network administrators. In that regard, network traffic classification offers a promising solution by precisely identifying IoT devices to enhance network visibility, allowing better network security. Currently, most IoT device identification solutions revolve around machine learning, outperforming prior solutions like port and behavioural-based. Although performant, these solutions often experience performance degradation over time due to statistical changes in the data. As a result, they require frequent retraining, which is computationally expensive. Therefore, this article aims to improve the model performance through a robust alternative feature set. The improved feature set leverages payload lengths to model the unique characteristics of IoT devices and remains stable over time. Besides that, this article utilizes the proposed feature set with Random Forest and OneVSRest to optimize the learning process, particularly concerning the easier addition of new IoT devices. On the other hand, this article introduces weekly dataset segmentation to ensure fair evaluation over different time frames. Evaluation on two datasets, a public dataset, IoT Traffic Traces, and a self-collected dataset, IoT-FSCIT, show that the proposed feature set maintained above 80% accuracy throughout all weeks on the IoT Traffic Traces dataset, outperforming selected benchmark studies while improving accuracy over time by +10.13% on the IoT-FSCIT dataset.

## INTRODUCTION

The Internet of Things (IoT) is a collection of electronic devices with sensors and software that aims to connect and exchange data, such as voice assistants, smart doorbells, and

smart locks. The extensive IoT systems present an opportunity to produce, collect, and process a tremendous amount of data for monitoring and decision-making purposes. The IoT data handling and processing capabilities benefit various industries, such as financial, healthcare, and automotive. For example, the financial sector implements IoT in contactless payment. Contactless payment allows consumers to complete payment without physical touch, which is especially beneficial during the COVID-19 pandemic when people need to minimize physical contact. Furthermore, Fortune Business Insights recently reported that the global IoT market is set to surpass USD 4 trillion by 2032, reflecting a Compound Annual Growth Rate (CAGR) of 24.3% over the forecasted period spanning from 2024 to 2032 (*Fortune Business Insights, 2024*).

Although set to soar in the near future, IoT's extensive systems present security vulnerabilities that require immediate attention, including malware attacks, Distributed Denial of Services (DDoS), and data theft. For example, Mirai, a malware attack designed to exploit IoT devices, caused a large-scale DDoS attack and successfully compromised over 600,000 connected IoT devices as of 2016 (*Tahaei et al., 2020*). With these security vulnerabilities, cyber attackers can mount attacks against the user's network through the exploited IoT devices. Potential solutions include IoT device identification, which maps network traffic to its originating IoT devices. Accurate and reliable IoT device identification allows IoT system administrators to identify IoT devices connected to their network and determine their vulnerability status, making it easier to address security concerns and prevent potential cyber-attacks. The absence of IoT device identification reduces the potential to identify network vulnerabilities accurately.

In recent years, there are various efforts from the research community to address IoT device identification. Based on the latest literature, there are at least three primary techniques in IoT device identification:

(a) *Signature-based*. It utilizes predefined rules or strings that allow highly accurate and granular device identification (*Zaki et al., 2021*). In addition to building the rules from scratch, existing works have also explored rule extraction from open-source tools like Nmap (*Wan et al., 2023*).

(b) *Traditional machine learning*. It is the most widely explored technique due to its effectiveness and the abundance of IoT data available for training. Among common algorithms include the k-nearest neighbour (k-NN), support vector machine (SVM) and Random Forest (*Almotairi et al., 2024*).

(c) *Deep learning*. The current state-of-the-art with advanced capabilities, such as automated feature representation and complex inferences through deep hidden layers. Well-known algorithms include transformer-based and convolutional neural networks (CNN) (*He et al., 2022*; *Luo et al., 2023*).

Implementing deep learning has emerged as an effective approach to identifying IoT devices. For instance, *He et al. (2022)* explored the integration of CNN with federated learning to process network traffic data efficiently, yielding significant improvements in device identification accuracy. Similarly, *Li et al. (2023)* demonstrated using CNN models to analyze channel state information (CSI), further substantiating the model's utility in identifying subtle patterns that distinguish devices. The research outlined in

*Liu et al. (2021b)* also delved into using CNNs for IoT security, highlighting their capacity to identify devices by analyzing temporal and spatial features within network signals. *Li et al. (2022)* further corroborated these findings by employing CNN to extract features from CSI data, providing a robust framework for IoT security. These studies collectively underscore the critical role of CNN in extracting intricate features from IoT network traffic, paving the way for more secure and reliable device identification strategies.

However, this article opted for Random Forest over the more advanced deep learning algorithms, which is justified by the results of two significant studies. The efficacy and relevance of conventional machine learning methods such as Random Forest in this area are underscored by the findings in *Hamad et al. (2019)* and *Kostas, Just & Lones (2022)*. These studies emphasized the algorithm's superiority in effectively recognizing devices by processing feature-rich network data, leveraging Random Forest's resistance to overfitting and its inherent ability to handle high-dimensional data without extensive computational resources. Furthermore, the meticulous features analysis and selection are in accordance with the strengths of Random Forest, underscoring the significance of a systematic approach to feature selection that improves model precision and generalizability. This article's choice for utilizing Random Forest is further supported by its established success in efficiently managing IoT devices' diverse characteristics and communication patterns. It provides a strong argument against the resource-heavy demands associated with deploying deep learning models in the IoT security domain.

Despite encouraging results of existing studies, existing machine learning-based IoT device identification systems face a common limitation: accuracy degradation over time. Accuracy degradation over time is a phenomenon in machine learning that occurs when the model's performance degrades due to the statistical properties between the training and testing data diverging over time, reducing the meaning and relationship between the data and making the model less suitable over time (*Iwashita & Papa, 2019*). The authors in *Kolcun et al. (2021)* highlighted that the model's accuracy decreased by up to 40% after a few weeks. Similarly, we extended the study in *Kolcun et al. (2021)* through empirical evaluation on the effect of different feature representations against the magnitude of accuracy degradation over time in our previous work (*Aqil et al., 2022*). One way to address the accuracy degradation over time in machine learning is by repeatedly retraining the model (*Kolcun et al., 2020*), a computationally expensive process requiring extensive human efforts. Despite its critical accuracy degradation, it remains an open research problem in the IoT domain.

To address the accuracy degradation in the IoT device identification system, this article proposes a flexible and robust feature set based on the statistical values of the payload length. The statistical values, such as average payload length and payload range, are unique per device, thus helping identify and differentiate one IoT device from another. This article incorporates the Random Forest algorithm and the OneVSRest strategy. OneVSRest is a strategy that fits one classifier per class, and each classifier is fitted against all other classes. The OneVSRest strategy is efficient for IoT systems by simplifying the data labeling process for a newly connected IoT device. In summary, the contributions of this article are as follows:

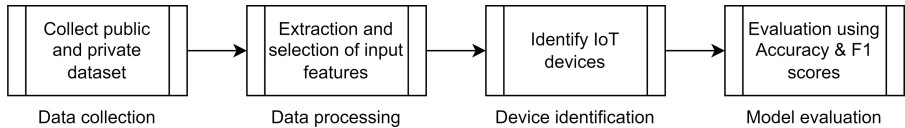

**Figure 1  IoT device identification phases.**

(a) This article offers a robust and lightweight feature set based on statistical values of payload length to identify and discriminate IoT devices while also addressing accuracy degradation over time to cater to IoT systems' complexity.

(b) This article provides a new and publicly available dataset as a new benchmark for IoT device identification.

This article also proposed an alternative evaluation strategy in which we split the dataset on a weekly basis to evaluate the accuracy degradation over time. The first week of the dataset was used for training purposes, and the following weeks were used for testing. The article used the public dataset, IoT Traffic Traces, from UNSW for benchmarking purposes with the benchmark article (*Kolcun et al., 2021*). The evaluation of the proposed feature set showed only a 6.8% accuracy degradation and 6.22% F1 score degradation while maintaining the accuracy rate at 83%. Compared to *Kolcun et al. (2021)*, the accuracy rate dropped as low as 78.37% in the sixth week, with slightly minimal degradation at 4.4%. In addition, this article collected a dataset, IoT-FSCIT (IoT Faculty Science Computer & Information Technology), over six weeks in a controlled lab environment, which includes five different IoT devices and contains bidirectional network traffic in packet captured format files (PCAP) format. Using a similar approach, the accuracy and F1 score degradation for the proposed feature set on this dataset are 10.81% and 12.81 respectively. Meanwhile, the benchmark article (*Kolcun et al., 2021*) demonstrated a larger percentage, at 20.94% for accuracy and 20.97 for the F1 score. In conclusion, the article showed that the proposed feature set maintains a higher performance level over time, even in different network environments.

Our article is organized as follows: 'Materials & Methods' presents the architecture of the proposed approach. 'Results' outlines the experimental analysis. 'Discussion' discusses the article's limitations and the current issue in the domain before concluding the article.

## MATERIALS & METHODS

This article identifies IoT devices using machine learning techniques and divides the process into four phases. Figure 1 illustrates the overall architecture of the approach, and the following subsections provide more details on each phase.

### Data collection phase

The data collection phase involves gathering relevant data to create a comprehensive dataset for analysis and modeling. Our article employs two datasets from different sources, as shown in Table 1, with their details as follows:

**Table 1  Details on the datasets used.**

| Dataset (Year) | Type | No. of devices | Devices |
|---|---|---|---|
| IoT Traffic Traces (2018) | Public | 23 IoT devices, seven non-IoT devices | Amazon Echo, Android Phone 1, Android Phone 2, Belkin Wemo Motion Sensor, Belkin Wemo Switch, Blipcare Blood Pressure Meter, Dropcam, HP Printer, iHome Power Plug, Insteon Camera, IoT Camera 1, iPhone, Laptop, Light Bulbs LiFX Smart Bulb, Macbook, iPhone, Nest Dropcam, Nest Protect Smoke Alarm, Netatmo Weather Station, Netatmo Welcome, PIX-STAR Photo Frame, Samsung Galaxy Tab, Samsung SmartCam, Smart Things, TP-Link Day Night Cloud Camera, TP-Link Smart Plug, Triby Speaker, Withings Aura Smart Sleep Sensor, Withings Smart Baby Monitor, Withings Smart Scale |
| IoT-FCSIT (2022) | Private | Five IoT devices | Alexa Echo Dot, Mi Air Purifier, Mi Box 3, Mi Home Security Camera, Smart Plug |

(a) IoT Traffic Traces: A publicly available dataset from the University of New South Wales (UNSW) (*Sivanathan et al., 2019*). The IoT Traffic Traces includes network traffic from 23 IoT devices and seven non-IoT devices, collected over 60 days. The PCAP files are segregated based on each day.

(b) IoT-FCSIT: Our dataset was collected over six weeks explicitly for this article's purpose of assessing the proposed feature set's robustness across various network environments. This dataset simulates the average consumer's typical usage, such as the Smart Plug being turned on when entering the lab and off when leaving. It was gathered over six weeks in the research lab at the Faculty of Computer Science and Information Technology (FCSIT), Universiti Malaya (UM), and is available to access through DOI: 10.6084/m9.figshare.25143581.v1.

## Data processing phase

The data processing phase involves preparing raw data for model training. The outcome is a processed dataset that is error-free, contains relevant features, and is appropriately structured for machine learning algorithms. The data processing phase encompasses four essential processes: data cleaning, feature selection, data transformation, and data splitting.

Figure 2 shows the data cleaning step where the raw data in a PCAP file is processed using the Pyshark library to transform it into a dataframe, which is then saved as a CSV file. Then, any data points with incomplete information are removed from the dataset to handle missing data. Additionally, non-IoT traffic, irrelevant to the current analysis, is eliminated from the dataset. Our article filters out non-IoT traffic based on MAC and IP addresses for the public dataset, IoT Traffic Traces. Meanwhile, for the private dataset, IoT-FSCIT, we configured our setup exclusively to ensure our dataset is free from any non-IoT traffic from the start, thereby eliminating the need for post-collection filtering.

The features implemented in this article were originally employed in our previous work on granular network traffic classification (*Zaki et al., 2022*). Our previous article aimed to classify network traffic based on inter-application and intra-application services. Inter-application services are similar services associated with different parent applications,

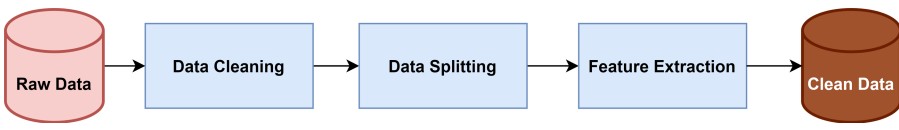

**Figure 2** Data processing.

**Table 2** 52 initial features.

| No. | Feature | Description |
| --- | --- | --- |
| 1–20 | paylod_mov_stat | 5, 10, 20, 40, and 100-moving statistics (*i.e.*, sum, average, variance, and standard deviation) of payload length. Each statistic is calculated on five different packet sliding windows. |
| 21 | max_avg_payload | The maximum average payload length in either traffic direction. |
| 22 | min_avg_payload | The minimum average payload length in either traffic direction. |
| 23–25 | avg_payload_per_second | Average payload length per 1, 5, and 10 s. |
| 26–30 | Range | The payload length range for the first 5, 10, 20, 40 and 100 packets, *i.e* maximum–minimum. |
| 31–45 | payload_first_stat | The statistic of payload length for the first 5, 10, 20, 40, and 100 packets (*i.e.*, sum, average, and standard deviation. |
| 46–50 | mss_count | The count of packets in the first 5, 10, 20, 40, and 100 packets have payload lengths equaling the maximum segment size. |
| 51 | ma_40_avg_50 | The average of the first five entries of the 40-packet moving average for payload length. |
| 52 | Protocol | The protocol of layer 4. |

like Facebook-comment and YouTube-comment. On the other hand, intra-application services encompass different services within the same parent application, for instance, YouTube-comment and YouTube-post. We introduced 52 initial payload-length-based features in the previous work, as documented in Table 2. We have extended the previous work by implementing these same features to effectively reduce accuracy degradation over time in IoT device identification systems.

Next, we compute Pearson's correlation coefficient ($r$) to determine the most relevant features. Pearson's correlation coefficient measures the statistical relationship and magnitude of the correlation between continuous variables. Features with low correlation values are less dependent on other variables and are thus more effective in diverse network environments. For this article, we selected features with only low degree ($-0.29 \leq r \leq 0.29$) and moderate degree (ranging from $-0.49 \leq r \leq -0.30$ and $0.30 \leq r \leq 0.49$) correlation values to boost their adaptability to different network environments. Considering the reliability of correlation values, we found that the feature max_payload consistently exhibits the lowest correlation values in both the public (IoT Traffic Traces) and our (IoT-FCSIT) datasets, which indicates its robustness in the context of Pearson correlation.

**Table 3  The seven selected features with Pearson correlation coefficient.**

| No. | Feature | Description | Pearson correlation coefficient | |
| --- | --- | --- | --- | --- |
| | | | Public | Private |
| 1 | avg_payload_1_second | The average payload length in 1 s duration. | −0.08192188 | −0.10889866 |
| 2 | fma5_sum | The sum of the 5-moving of payload length. | −0.0444108 | −0.09048064 |
| 3 | fma5_mean | The mean of the 5-moving of payload length. | −0.04437536 | −0.09047919 |
| 4 | fma5_variance | The variance of the 5-moving of payload length. | −0.19901667 | −0.08384815 |
| 5 | fma5_std | The standard deviation of the 5-moving of payload length. | −0.16614413 | −0.08440375 |
| 6 | range_5 | The payload length range for the first five packets, *i.e.,* maximum–minimum. | −0.16258816 | −0.08955199 |
| 7 | max_payload | The maximum payload length in either traffic direction. | −0.33862645 | −0.41554773 |

Additionally, instead of using 10, 20, 40, and 100 packets to enhance efficiency, we opt for a more efficient window size of five packets. A smaller window size allows quicker updates to the statistical feature values, making the feature set more responsive to changes in network traffic patterns. These findings contribute to developing an efficient IoT device identification model while reducing accuracy degradation over time. Table 3 lists the seven selected features, including each feature's description and correlation coefficients.

The following equations describe all the selected features using statistical measures from the payload length. These measures help us gain insights into the characteristics of the network traffic.

*Average payload length within a 1-second duration (avg_payload_1_sec)*

The first feature, *avg_payload_1_sec*, quantifies the average payload length across a discrete 1-second interval. It entails aggregating the payload lengths of all packets within this temporal window and subsequently dividing this sum by the total count of packets (denoted as $n$) within this interval. Equation (1) shows the definition:

$$\text{avg}_{\text{payload}_{1_{\text{sec}}}} = \frac{1}{n} \sum_{j=1}^{n} P_j \tag{1}$$

where, $n$ signifies the total number of packets within the 1-second interval, and $P_j$ corresponds to the payload length of the $j$-th packet. Utilizing the 1-second duration is preferred due to its minimal time frame, as opposed to five or 10-second intervals. This choice ensures a more rapid response to changes in network traffic and expedites the feature extraction process.

*Statistical measures employing a 5-moving sliding window.*

This suite of features (*i.e.,* sum, mean, variance, and standard deviation) leverages a 5-moving sliding window technique to analyze the payload lengths. Let $P_i$ denote the payload length of the $i$-th packet and the 5-moving sliding window is computed for each packet from the current packet, $i$, to the four subsequent packets ($i+4$). Equation (2) shows the summation of payload lengths within a 5-moving sliding window:

$$\text{fma5}_{\text{sum}} = \sum_{j=1}^{5} P_i. \tag{2}$$

The mean payload length within this window is calculated by dividing the sum of the payload lengths by the window size, as shown in Eq. (3):

$$\text{fma5}_{\text{mean}} = \frac{1}{5}\text{fma5}_{\text{sum}i}. \tag{3}$$

The variance, indicative of data dispersion, is ascertained by computing the average squared difference between each payload length and the mean within the window, denoted in Eq. (4) as *fma5_variance*:

$$\text{fma5}_{\text{variance}} = \frac{1}{5}\sum_{j=1}^{5}\left(P_{i+j} - \text{fma5}_{\text{mean}i}\right)^2. \tag{4}$$

Then, the standard deviation, reflecting data scatter, is the square root of the computed variance, as shown in Eq. (5):

$$\text{fma5}_{\text{std}} = \sqrt{\text{fma5}_{\text{variance}i}}. \tag{5}$$

*Payload length range for the initial five packets (range_5)*
This feature gauges the range between the maximum and minimum payload lengths among the first five packets of a traffic flow:

$$\text{range}_5 = \max\left(\sum_{i=1}^{5}P_i\right) - \min\left(\sum_{i=1}^{5}P_i\right). \tag{6}$$

*Maximum payload length in either traffic direction (max_payload)*
The last feature evaluates the maximal payload length between two traffic directions: source to destination ($\text{max}_a$) and destination to source ($\text{max}_b$). The formulation is as follows:

$$\text{max}_{\text{payload}} = \text{maximum}(\text{max}_a, \text{max}_b). \tag{7}$$

where $\text{max}_a$ signifies the average payload length computed from the source to the destination within a given traffic flow, and $\text{max}_b$ represents the average payload length calculated from the destination to the source for the same traffic flow.

After completing the feature preparation, the processed dataset includes the necessary labels for classification. To make the labels suitable for machine learning algorithms, they are transformed into an appropriate categorical representation using the Label Encoder provided by the Scikit-learn library. This transformation process is essential as it converts categorical features into a suitable numerical representation, ensuring the data is well-prepared for machine learning algorithms.

Furthermore, unlike other existing studies (*Charyyev & Gunes, 2021*; *Liu et al., 2021a*; *Najari et al., 2020*) that divide their dataset into training and testing data, this article follows the partitioning strategy as in (*Kolcun et al., 2021*) to evaluate accuracy degradation over time, which partitioned its datasets into several weeks intervals. However, we partitioned the datasets into one-week intervals due to the duration covered by the datasets utilized in our article—spanning six and eight weeks. The training set comprises data from the initial week (Week 1), while the testing set consists of data from subsequent weeks (Week 2 to

**Table 4   Parameters value for the Random Forest strategy.**

| No. | Parameters | Description | Value |
|---|---|---|---|
| 1. | *n*_estimators | The number of trees in the forest | 100 |
| 2. | max_depth | The maximum number of trees | None |
| 3. | min_samples_split | The minimum number of samples required to split an internal node | 10 |
| 4. | min_samples_leaf | The minimum number of samples required to be at a leaf node | 2 |
| 5. | max_features | The number of features to consider when looking for the best split | Auto |
| 6. | max_leaf_nodes | Grow trees | None |
| 7. | random_state | Controls both the randomness of the bootstrapping of the samples used when building trees and the sampling of features to consider when looking for the best split each node | 0 |

Week 8). This strategic division of datasets allows for a comprehensive assessment of the model's performance over an extended period, ensuring its efficacy in handling IoT device identification with improved accuracy.

## Identification phase

This article introduces a hybrid identification technique that combines the following approaches as follows:

(a) *Random Forest algorithm*. Widely adopted in IoT device identification models, Random Forest is a supervised machine learning method renowned for its high accuracy and efficiency (*Hamad et al., 2019*). It achieves its accuracy and effectiveness by aggregating predictions from multiple decision trees, effectively reducing the risk of overfitting.

(b) *OneVsRest strategy*. This strategy constructs individual classifiers for each class and trains them against all other classes (*Miettinen et al., 2017*). Doing so simplifies the labelling process for new IoT devices and enables a detailed analysis of each class by examining its respective classifiers.

The hybrid approach leverages the strength of the Random Forest algorithm's ensemble method and the versatility of the OneVsRest strategy to achieve accurate and efficient IoT device identification. The parameters listed in Table 4 play a crucial role in optimizing the Random Forest algorithm's performance, ensuring its robustness and effectiveness in handling diverse IoT environments.

## Evaluation phase

The evaluation phase involves evaluating the machine learning model's performance using various metrics. This evaluation process yields valuable insights into the model's effectiveness. The metrics employed in this article are as follows:

(a) *Accuracy*. The accuracy gauges the overall correctness of the model's predictions. It quantifies the proportion of correctly classified instances out of the total instances in the

dataset. The accuracy is calculated using the formula in Eq. (8):

$$\text{Accuracy} = \frac{\text{TP} + \text{TN}}{\text{TP} + \text{FP} + \text{TN} + \text{FN}}. \tag{8}$$

(b) *F1 score*. The F1 score offers a comprehensive evaluation, especially in imbalanced datasets, ensuring a more reliable assessment of the IoT device identification system's performance. The F1 score is calculated using the formula in Eq. (9):

$$F1\text{score} = 2 \times \frac{\text{Precision} \times \text{Recall}}{\text{Precision} + \text{Recall}}. \tag{9}$$

The definitions for the components of these formulas are as follows:

- *True positive (TP)*. The instances where the model correctly identifies device A as device A and device B as device B.
- *True negative (TN)*. The instances where the model correctly identifies data that is not device A as not device A and data that is not device B as not device B.
- *False positive (FP)*. Represents false alarms or incorrect positive predictions.
- *False negative (FN)*. Represents missed positive predictions.
- *Precision*. Measures the proportion of actual positive instances the model correctly identifies.
- *Recall*. Measures the proportion of instances the model correctly identified as positive out of all predicted instances.

These metrics enable a comprehensive evaluation of the model's performance, facilitating a robust analysis of the IoT device identification system's capabilities.

## RESULTS

Our article conducted multiple experiments to evaluate the proposed approach's capability to reduce accuracy degradation over time. We implemented the proposed approach on the public dataset (IoT Traffic Traces) to facilitate benchmarking with existing approaches. Additionally, we applied the proposed approach to our dataset (IoT-FCSIT), which simulates the daily routine of an average user, to showcase its robustness in different network environments. Three key aspects explored are accuracy and F1 score degradation and technical analysis on the cause of accuracy degradation.

### Accuracy & F1 scores degradation over time

In this section, we present the accuracy and F1 scores degradation analysis for both the proposed feature set and the flow-based feature set on the IoT Traffic Traces and IoT-FSCIT datasets in Tables 5 and 6, respectively. The experimental results for each dataset are summarised below:

 *Accuracy scores on public dataset (IoT Traffic Traces):*

(a) The proposed feature set initially achieved an accuracy of 89.13%, gradually declining over eight weeks to its lowest point of 83.05%.
(b) Throughout the two-month testing period, the proposed feature set consistently maintained an accuracy level above 80%, with an overall degradation of 6.8%.

**Table 5 The accuracy scores on the public dataset.**

| Week | Accuracy (%) | |
|---|---|---|
| | **Proposed features** | **Flow-based** |
| 1 | 89.13 | 82.44 |
| 2 | 84.62 | 80.98 |
| 3 | 86.41 | 79.90 |
| 4 | 83.33 | 78.77 |
| 5 | 84.71 | 82.69 |
| 6 | 83.67 | 78.01 |
| 7 | 85.24 | 78.47 |
| 8 | 83.05 | 79.53 |
| Average | 85.02 | 80.10 |

**Table 6 The accuracy scores on the private dataset.**

| Week | Accuracy (%) | |
|---|---|---|
| | **Proposed features** | **Flow-based** |
| 1 | 99.50 | 99.83 |
| 2 | 95.96 | 96.45 |
| 3 | 97.81 | 88.72 |
| 4 | 96.73 | 79.28 |
| 5 | 97.06 | 89.25 |
| 6 | 89.69 | 78.89 |
| Average | 96.13 | 88.73 |

(c)  In contrast, the flow-based feature set exhibited lower accuracy scores compared to the proposed feature set throughout the testing period.

(d)  The flow-based feature set consistently dropped below 80% accuracy on most weeks, indicating a less reliable performance.

(e)  Despite the lower degradation observed in the flow-based feature set, the proposed feature set outperformed it in terms of overall accuracy scores, maintaining above 80% accuracy on all weeks.

(f)  Moreover, the proposed feature set achieved higher accuracy while utilising less than half of the features proposed in a previous article, aligning with the objective of proposing a robust feature set against accuracy degradation over time.

*Accuracy scores on our dataset (IoT-FCSIT):*

(a)  Proposed feature set: Initial accuracy of 99.5%, final accuracy of 89.69%, and degradation of 10.81% over the evaluation period.

(b)  Flow-based feature set: Initial accuracy of 99.83%, lowest accuracy of 78.89% (Week 4), and degradation of 20.94% over the evaluation period.

(c)  The proposed feature set consistently outperformed the flow-based feature set in accurately classifying devices within the IoT-FSCIT dataset.

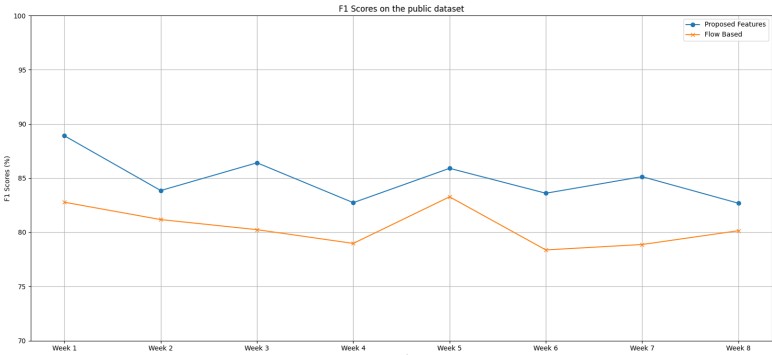

**Figure 3** **The F1 scores degradation over weeks on the public dataset.**

(d) The results of the comparison show that the proposed feature set is superior in terms of accuracy degradation, demonstrating its robustness against accuracy deterioration in different network environments.

(e) Additionally, the proposed feature set achieved higher accuracy while utilising fewer features than in a previous article, highlighting the efficiency and effectiveness of the selected features.

(f) Both approaches achieved higher accuracy scores in the IoT-FSCIT dataset compared to the IoT Traffic Traces dataset, attributed to the reduced complexity and smaller number of IoT devices in the IoT-FSCIT dataset.

In conclusion, the accuracy degradation analysis on both datasets demonstrates the superior performance of the proposed feature set over the flow-based feature set. The proposed feature set showcases robustness against accuracy degradation, highlighting its potential for real-world applications in diverse IoT environments. The findings are further analysed in the next section.

*F1 scores on public dataset (IoT Traffic Traces):*

We notice distinct trends in the performance of both the proposed feature set and the flow-based feature set when evaluating the IoT Traffic Traces dataset over an eight-week testing period, as shown in Fig. 3, as follows:

(a) During the first week of testing, both methods exhibited remarkable performance, with the proposed feature set reaching an F1 score peak of 88.90% compared to the flow-based feature set's lower score of 82.77%.

(b) Week 2 saw a slight dip in the F1 scores of both methods. The proposed feature set came in at 83.85%, and the flow-based feature set wasn't far behind at 81.17%.

(c) From Week 3 through Week 6, the proposed feature set displayed a competitive edge, consistently scoring higher F1 scores than the flow-based method.

(d) The gap between the two methods widened significantly during Week 6, with the proposed feature set achieving an F1 score of 83.60%, overshadowing the flow-based feature set's score of 78.37%.

(e) In the final weeks, we see both models improving but maintaining the performance gap. This observation hints at the robustness of the proposed feature set.

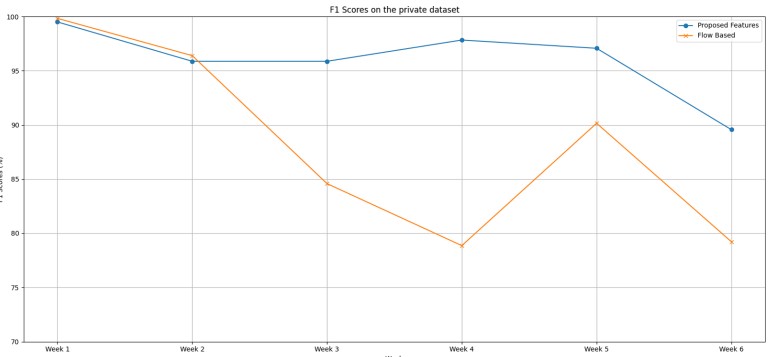

**Figure 4** The F1 scores degradation over weeks on the private dataset.

(f) The proposed feature set holds its lead with an average F1 score of 84.90%, while the flow-based feature set trails with an average of 80.48%.

*F1 scores on our dataset (IoT-FCSIT):*

Following are the trends of the proposed features set and flow-based feature set over a six-week testing period, as shown in Fig. 4:

(a) In the initial week, both feature sets demonstrate impressive performance. The proposed feature set achieves a near-perfect F1 score of 99.5%, while the flow-based feature set surpasses it slightly with an F1 score of 99.83%.

(b) The second week sees a minor performance drop in both feature sets. Despite this dip, the flow-based feature set continues to hold a slight edge.

(c) From Week 3 onwards, a dramatic shift occurs. The proposed feature set consistently outperforms the flow-based feature set up until Week 6. The disparity between the two feature sets is most pronounced in Week 4.

(d) Over the entire period, the proposed feature set outshines the flow-based feature set with an average F1 score of 96.09% compared to the flow-based feature set's average of 88.17%.

(e) This substantial difference of approximately 7.92% in favour of the proposed feature set is not trivial. It highlights the superior predictive capability and stability of the proposed feature set over time, even when faced with datasets extracted from different network environments.

In conclusion, the F1 score degradation analysis on both datasets reiterates the robustness and superiority of the proposed feature set over the flow-based feature set. The findings highlight the effectiveness and stability of the proposed features in accurately classifying IoT devices over time, contributing to enhanced IoT traffic analysis in various network environments. Additionally, evaluating the IoT-FSCIT dataset showcases the adaptability of the proposed features, yielding improved accuracy and performance in a reduced complexity setting. These results further underscore the importance of selecting and refining features to address accuracy degradation and concept drift challenges in machine learning applications.

## DISCUSSION

This section presents a comprehensive analysis of accuracy degradation on both IoT Traffic Traces and IoT-FSCIT datasets using the proposed feature set. We evaluate their performance using several tools, such as a confusion matrix, plotting the same feature's values of the same devices, and the Kolgomorov-Smirnov test. We split our technical analyses into two categories: misclassification among streaming devices and misclassification among devices of the same brand.

### Misclassification among streaming devices

We ran the confusion matrix to provide better visualisation of misclassifications in both approaches on both datasets. The confusion matrix is a performance evaluation tool in machine learning to summarise the predictions made by the classification model and gain insights into its performance. The confusion matrix analysis on both the IoT Traffic Traces and IoT-FSCIT datasets revealed notable misclassifications among streaming devices. Complex data transmission processes and diverse network behaviours characterise streaming devices.

On the IoT Traffic Traces dataset, we ran the confusion matrix on Week 8 of the flow-based features set, as depicted in Table 7. The findings are as follows:

(a) Amazon Echo: Misclassified as Belkin Wemo Switch 2.25%, Belkin Wemo Motion Sensor 2.52%, and HP Printer 1.75%.

(b) HP Printer: Misclassified as Amazon Echo 0.41%, as Belkin Wemo Switch 5.58%, Belkin Wemo Motion Sensor 0.75%, Insteon Camera 2.01%, Samsung SmartCam 0.76%, and as Triby Speaker 0.93%.

(c) Insteon Camera: Misclassified as Belkin Wemo Switch 0.42%, Belkin Wemo Motion Sensor 3.33%, and Samsung SmartCam 0.20%.

(d) Samsung SmartCam: Misclassified as Belkin Wemo Switch 0.01% and Insteon Camera 0.15%.

(e) Triby Speaker: Misclassified as Belkin Wemo Switch 0.06% and Insteon Camera 0.10%.

Table 8, the confusion matrix on Week 6 for proposed features set on IoT-FSCIT dataset also reveals misclassifications among streaming devices, which include Alexa Echo Dot, Mi Box 3, and Mi Home Security Camera. The findings are as follows:

(a) Alexa Echo Dot: Misclassified as Mi Box 3 3.6% and as Mi Home Security Camera 0.7%

(b) Mi Box 3: Misclassified as Alexa Echo Dot 6.42% and as Mi Home Security Camera 2.55%.

(c) Mi Home Security Camera: Misclassified as Alexa Echo Dot 0.51% and as Mi Box 3 0.43%.

The confusion matrix analysis on both the IoT Traffic Traces and IoT-FSCIT datasets revealed significant misclassifications among streaming devices, indicating challenges in accurately identifying such devices. On the IoT Traffic Traces dataset, the flow-based features set displayed misclassifications primarily among Amazon Echo, HP Printer, Insteon Camera, Samsung SmartCam, and Triby Speaker. These devices were frequently misclassified as other streaming devices, suggesting similar network traffic patterns.

Aqil et al. (2024), *PeerJ Comput. Sci.*, DOI 10.7717/peerj-cs.2145

**Table 7  The confusion matrix of week eight for the flow-based features set.**

| Device | Amazon echo | Belkin wemo switch | Belkin Wemo Motion Sensor | HP printer | Insteon camera | Lifx smart bulb | Nest protect smoke alarm | Netatmo weather station | Samsung smartcam | Smart things | Triby speaker | Withings smart scale |
|---|---|---|---|---|---|---|---|---|---|---|---|---|
| Amazon Echo | 243276 | 480 | 6132 | 4296 | 5 | 0 | 0 | 0 | 344 | 0 | 0 | 0 |
| Belkin Wemo Switch | 0 | 255469 | 39335 | 3 | 763 | 0 | 0 | 0 | 4995 | 0 | 9071 | 0 |
| Belkin Wemo Motion Sensor | 0 | 124643 | 414730 | 53 | 56662 | 0 | 0 | 0 | 37398 | 0 | 36935 | 0 |
| HP Printer | 335 | 4574 | 6194 | 16261 | 2053 | 0 | 0 | 0 | 7615 | 0 | 9293 | 0 |
| Insteon Camera | 0 | 1095 | 8626 | 38 | 256230 | 0 | 0 | 0 | 2512 | 0 | 486 | 0 |
| Lifx Smart Bulb | 0 | 0 | 0 | 1 | 3 | 75760 | 1 | 0 | 31 | 0 | 1 | 0 |
| Nest Protect Smoke Alarm | 0 | 0 | 0 | 0 | 0 | 2 | 1323 | 0 | 0 | 0 | 0 | 0 |
| Netatmo Weather Station | 0 | 0 | 0 | 0 | 0 | 0 | 0 | 49142 | 0 | 0 | 0 | 0 |
| Samsung Smartcam | 0 | 21 | 2079 | 5 | 427 | 0 | 0 | 0 | 291657 | 0 | 307 | 0 |
| Smart Things | 0 | 0 | 0 | 0 | 0 | 0 | 0 | 0 | 0 | 177137 | 0 | 0 |
| Triby Speaker | 0 | 135 | 234 | 1 | 96 | 0 | 0 | 0 | 330 | 0 | 40105 | 0 |
| Withings Smart Scale | 0 | 0 | 0 | 0 | 13 | 0 | 0 | 0 | 0 | 0 | 0 | 1776 |

**Table 8** The confusion matrix of week six for the proposed feature set.

| Device | Alexa echo dot | Mi air purifier | Mi box 3 | Mi home security camera | Smart plug |
|---|---|---|---|---|---|
| Alexa Echo Dot | 302731 | 0 | 11382 | 2173 | 0 |
| Mi Air Purifier | 0 | 87266 | 0 | 0 | 0 |
| Mi Box 3 | 35050 | 0 | 496600 | 13915 | 0 |
| Mi Home Security Camera | 2430 | 0 | 2041 | 474913 | 0 |
| Smart Plug | 22449 | 61137 | 20848 | 8559 | 203470 |

Similarly, the proposed feature set on the IoT-FSCIT dataset exhibited misclassifications among streaming devices, including Alexa Echo Dot, Mi Box 3, and Mi Home Security Camera.

Additionally, we can observe the accurate and high scores achieved for devices like LiFX Smart Bulb, NEST Protect Smoke Alarm, NETATMO Weather Stations, Smart Things, and Withings Smart Scale on the IoT Traffic Traces dataset and Mi Air Purifier on the IoT-FSCIT dataset. These findings emphasize the relative ease of correctly identifying non-streaming devices with straightforward network traffic patterns.

As the confusion matrix in Table 7 shows, HP Printer, categorised as one of the streaming devices, exhibited the lowest accuracy score of 35.1%. To investigate the underlying cause of this degradation, we conducted a Kolgomorov-Smirnov (KS) test on the 'Bidirectional Mean of Packet Size' and 'Source to Destination Bytes' features for HP Printer between Week 1 and subsequent weeks (Weeks 2 to 8) in the flow-based approach. The KS test results for both features are visualised in Figs. 5 and 6, respectively. The KS Test assesses the similarity between two distributions, with a small $p$-value indicating a significant difference and a larger $p$-value suggesting similarity, where the significance level is set to 0.05. The findings revealed the following:

(a)  Bidirectional Mean of Packet Size feature:
  ○  Week 2 *vs.* Week 1: The $p$-value is 0.0529, relatively close to significance. The distributions are considered the same, as the $p$-value is insignificant
  ○  Weeks 3 to 8 *vs.* Week 1: The $p$-values for Weeks 3 to 8 are extremely small, ranging from 1.347e−16 to 1.124e−78. This indicates the distributions of the 'Bidirectional Mean of Packet Size' feature for HP Printer significantly differ from Week 1 in all subsequent weeks.

(b)  Source to Destination Bytes feature:
  ○  Week 2 to Week 8 *vs.* Week 1: The $p$-values for all testing weeks, Weeks 2 to 8, are extremely small, ranging from 1.69e−4 to 2.11e−81. This indicates the distributions of the 'Source to Destination' feature for the HP Printer in all testings weeks differ from its training data, Week 1.

The confusion matrix in Table 8 shows that Smart Plug has big misclassifications, with only 64.3%. We again ran the KS Test on all seven features of the proposed features set. The results consistently show a $p$-value of 0.0 between Week 1 and subsequent weeks (Week 2 to 6), indicating that the distributions of these features for the Smart Plug are significantly different from Week 1. These significant differences in data distribution suggest that the

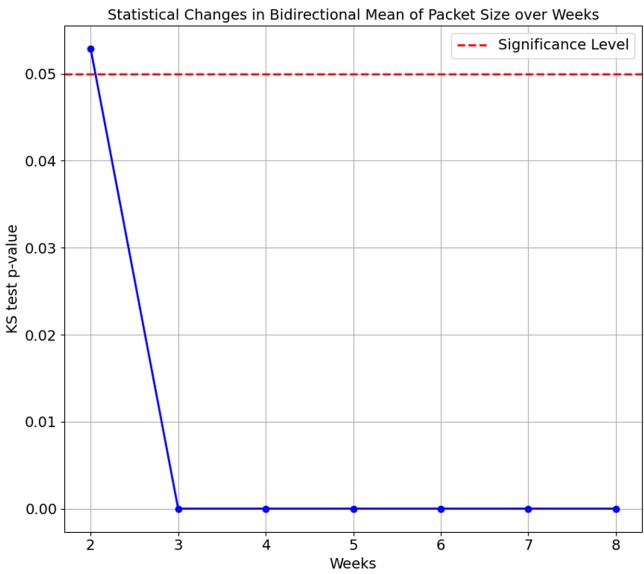

**Figure 5** Kolmogorov-Smirnov test of Bidirectional Mean of Packet Size feature over weeks.

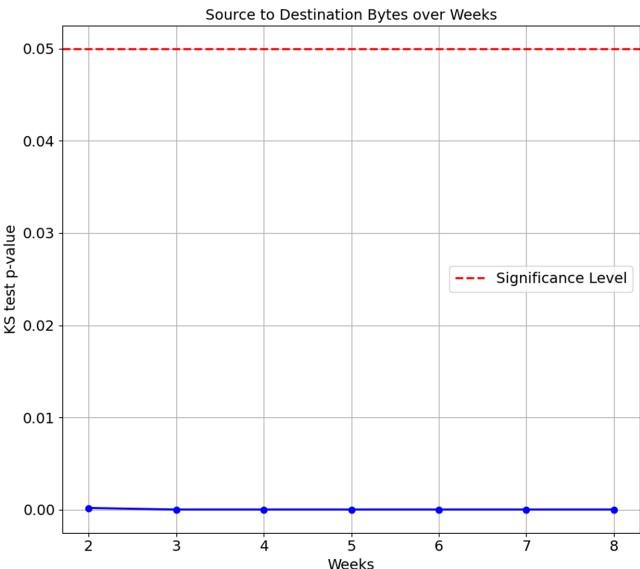

**Figure 6** Kolmogorov-Smirnov test of Source to Destination Bytes feature over weeks.

network behaviour of the Smart Plug has undergone substantial changes over time, which explain the big misclassifications observed in Table 8. Figures 7 and 8 illustrate two of the seven proposed features.

In summary, the statistical changes observed over time in the feature further corroborate the accuracy degradation findings, as the differences in distributions become more pronounced as time progresses. The model performance's degradation is observed as early

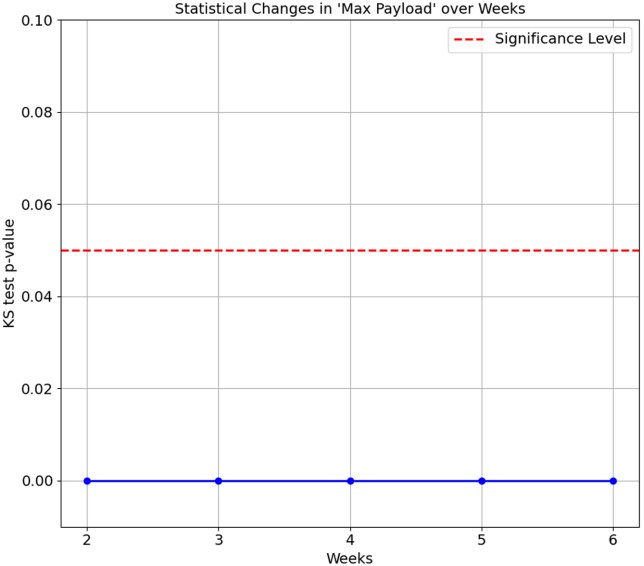

**Figure 7** Kolmogorov-Smirnov test for maximum payload over weeks.

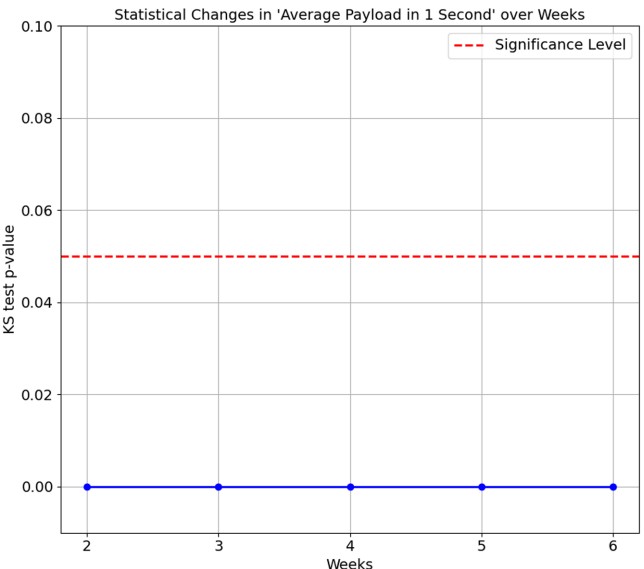

**Figure 8** Kolmogorov-Smirnov test for average payload in 1 second over weeks.

as Week 2, as (*Wan et al., 2023*) suggests. This degradation highlights a common problem in machine learning: the performance of a model degrades when there are significant changes in the data distribution over time. Maintaining model accuracy and reliability becomes particularly challenging in IoT environments, where device behaviour and network conditions evolve dynamically. Addressing these statistical changes accordingly is

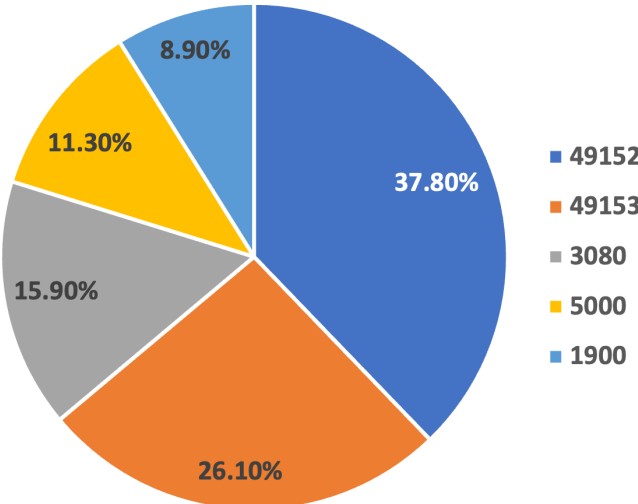

**Figure 9** **Top five most used destination ports for Belkin Wemo Motion Sensor.**

the main objective of this article, in which our proposed feature set managed to maintain above 80% accuracy scores on all weeks on both datasets.

## Misclassification among devices of the same brand

The confusion matrix also showcases the misclassification between devices from the same manufacturer or brand, such as the Belkin Wemo Switch and the Belkin Wemo Motion Sensor. This article found similar network signatures causing incorrect classifications.

(a) We plot the pie charts showing the top five most used destination ports for both devices, further demonstrating the similarity in their network traffic patterns, as shown in Figs. 9 and 10.

(b) The findings highlight the superior performance of the proposed approach, maintaining above 80% accuracy despite also getting caught up in confusion between Belkin devices.

## Experimental analysis and technical findings summary

This section provides a comprehensive analysis of accuracy degradation on the IoT Traffic Traces and IoT-FSCIT datasets by evaluating the performance of the proposed feature set using various tools, including confusion matrix, feature value plotting, and the Kolgomorov-Smirnov test. The findings summarisation are as follows:

(a) Confusion matrix analysis revealed significant misclassifications among streaming devices on both datasets, indicating difficulties in accurately identifying such devices.

(b) Non-streaming devices like LiFX Smart Bulb and NEST Protect Smoke Alarm achieved high and accurate scores on the IoT Traffic Traces dataset, suggesting their straightforward network traffic patterns were easier to classify.

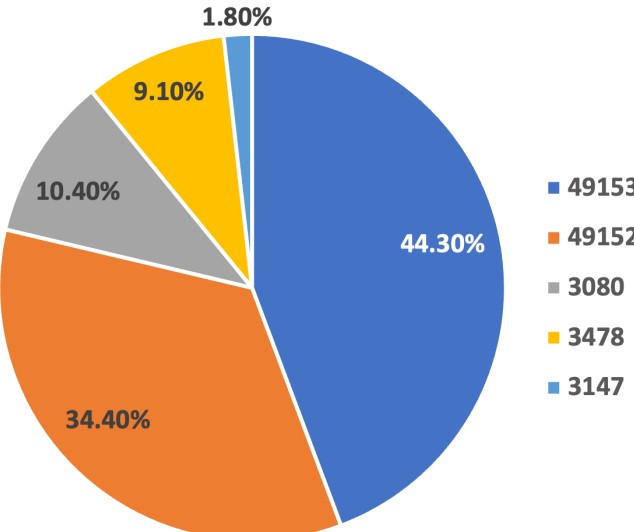

**Figure 10** Top five most used destination ports for Belkin Wemo Switch.

(c) HP Printer, classified as a streaming device, exhibited the lowest accuracy score of 35.1% on the IoT Traffic Traces dataset, pointing to challenges in maintaining accuracy in dynamic IoT environments.

(d) The Kolgomorov-Smirnov test showed substantial changes in data distribution over time for HP Printer's features, further highlighting the impact of dynamic IoT conditions on model performance.

(e) Despite misclassifications between devices from the same brand, such as Belkin Wemo Switch and Belkin Wemo Motion Sensor, the proposed feature set maintained an overall accuracy above 80%.

The technical analysis underscored the significant challenges in accurately classifying streaming devices, particularly due to their complex data transmission processes and diverse network behaviours. However, the proposed feature set demonstrated promising results, maintaining accuracy above 80% on both datasets. To enhance classification performance further, addressing statistical changes in data distribution over time and refining the model's ability to distinguish between devices from the same brand are crucial areas of focus in dynamic IoT environments.

## CONCLUSIONS

This article aimed to devise an approach that can reduce the accuracy degradation over time in IoT device identification while providing a more lightweight yet reliable solution. Our methodology and proposed feature set have been rigorously tested on two separate datasets. Our findings demonstrated that the proposed approach consistently maintained high accuracy levels, surpassing 80% on the IoT Traffic Traces dataset and nearing 90%

on the IoT-FCSIT dataset, even when utilizing a smaller feature set. This performance notably exceeded (*Kolcun et al., 2021*), which struggled to maintain similar thresholds. Nonetheless, our article's limitations include the ongoing issue with classifying streaming devices. To address this issue, we aim to refine the model further in our future work and explore additional features that can better capture their unique characteristics. Additionally, encompassing a broader range of devices is a viable path to enhance our approach's overall performance and applicability. We also acknowledge existing open issues, such as the significant challenge of developing and adapting models to new devices, changing device behaviour, and eliminating extensive retraining. However, our proposed approach, which utilized the OneVsRest strategy, is a step in the right direction to address such issues while exploring future novel approaches. In conclusion, our work has made notable strides in addressing the key limitations in IoT device identification, delivering a more robust feature set. We successfully demonstrated that maintaining high accuracy over time is indeed possible. We also successfully demonstrated that features from our previous work (*Zaki et al., 2022*) remain relevant and robust against accuracy degradation over time in the IoT device identification domain. Hence, the findings suggest that our methodology holds potential for applications in other domains beyond IoT device identification.

### Funding

This work was supported by Konsortium Kecemerlangan Penyelidikan (JPT(BKPI)1000/016/018/25 (49)) and the Fundamental Research Grant Scheme (FRGS/1/2023/ICT11/UM/02/1) provided by the Ministry of Higher Education of Malaysia. The funders had no role in study design, data collection and analysis, decision to publish, or preparation of the manuscript.

### Grant Disclosures

The following grant information was disclosed by the authors:
Konsortium Kecemerlangan Penyelidikan: (JPT(BKPI)1000/016/018/25 (49)).
Fundamental Research Grant Scheme: FRGS/1/2023/ICT11/UM/02/1.

### Competing Interests

The authors declare there are no competing interests.

### Author Contributions

- Nik Aqil conceived and designed the experiments, performed the experiments, analyzed the data, performed the computation work, prepared figures and/or tables, and approved the final draft.
- Faiz Zaki analyzed the data, authored or reviewed drafts of the article, grant Funding, and approved the final draft.
- Firdaus Afifi conceived and designed the experiments, analyzed the data, performed the computation work, prepared figures and/or tables, and approved the final draft.
- Hazim Hanif performed the computation work, authored or reviewed drafts of the article, and approved the final draft.

- Miss Laiha Mat Kiah analyzed the data, authored or reviewed drafts of the article, and approved the final draft.
- Nor Badrul Anuar conceived and designed the experiments, authored or reviewed drafts of the article, grant Funding, and approved the final draft.

## Data Availability

The raw data is available at figshare: Aqil (2024). IoT-FSCIT. figshare. Dataset. https://doi.org/10.6084/m9.figshare.25143581.v1.

The code is available in the Supplemental File.

## Supplemental Information

Supplemental information for this article can be found online at http://dx.doi.org/10.7717/peerj-cs.2145#supplemental-information.

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
