# Peer review of "Improved temporal IoT device identification using robust statistical features"

_PeerJ Computer Science, doi:10.7717/peerj-cs.2145_

## Round 0.1 · original submission · Minor Revisions

Please review the attached comments from the reviewers, make the changes based on their feedback, and submit your revised document by the deadline.

**Language Note:** The review process has identified that the English language must be improved. PeerJ can provide language editing services - please contact us at [email protected] for pricing (be sure to provide your manuscript number and title). Alternatively, you should make your own arrangements to improve the language quality and provide details in your response letter. – PeerJ Staff

·

Basic reporting

This study aims to reduce the accuracy degradation over time in IoT device identification.
The authors utilized two datasets to improve the accuracy over time in the IoT device identification system by proposing a statistical feature set based on payload length.

The proposed statistical feature set is employed in the Random Forest learning algorithm that is further incorporated with the scalable OneVSRest strategy for the data labeling process.

In my point of view, I feel that the paper is too long, some times too many details are presented while other subsections lack important details.

The paper needs a complete review, the author should consider typos and incomplete sentences for example
182 The seven selected features are listed below in?? Where are the selected features?

The abstract and conclusion subsections require some slimming down.

Experimental design

154 How do you locate non-IoT traffic, irrelevant to the current analysis?
211 Consider revising “The mean payload length within this window is calculated as the division of the sum five”

Validity of the findings

ok

Additional comments

ok

Reviewer 2 ·

Basic reporting

The study presents a significant advancement in the field of IoT device identification, employing a novel statistical feature set and machine learning techniques to address the challenge of accuracy degradation over time. However, like any research, it has its limitations and areas for potential improvement.

Overall, the paper needs improvements in several aspects, as explained below:

1. The literature review of this article is very terse. Many studies discussing the use of different types of feature extraction have not been discussed. The authors could improve this part by discussing the following studies: Robust event-based non-intrusive appliance recognition using multi-scale wavelet packet tree and ensemble bagging tree; Enhanced Feature Optimization for Multiclasss Intrusion Detection in IOT Fog Computing Environments; Effective non-intrusive load monitoring of buildings based on a novel multi-descriptor fusion with dimensionality reduction; Enhancing intrusion detection in IoT networks using machine learning-based feature selection and ensemble models

Experimental design

2. In addition to statical features, transforming one dimensional signals into 2D spaces and using CNN models to extract robust features can be explored in this study or at least discussed in the Introduction/literature review. More details can find in the following studies: An innovative deep anomaly detection of building energy consumption using energy time-series images; Using embedded feature selection and CNN for classification on CCD-INID-V1—a new IoT dataset; From time-series to 2d images for building occupancy prediction using deep transfer learning; Iot device identification using deep learning;

3. The significant misclassification among streaming devices indicates a challenge in accurately identifying devices with complex data transmission processes and diverse network behaviors. This suggests a need for further refinement in the feature set or the exploration of alternative features that can capture the unique characteristics of these devices more effectively.

Validity of the findings

4. The study acknowledges the issue of data divergence over time but does not fully address how to continually adapt the model without extensive retraining. While the proposed feature set shows promise, the methodology for updating the model to handle new devices or changes in device behavior over time could be further explored.

5. While the study demonstrates success in the datasets used, it's unclear how well the approach would generalize to other IoT environments with different device types or network configurations. Additional research in diverse settings could help validate the approach's applicability across the broader IoT domain.

Additional comments

6. Although the study aims to provide a more lightweight solution, the computational resources required for model training, especially in a real-world scenario with potentially thousands of IoT devices, are not fully addressed. The balance between model complexity, accuracy, and computational efficiency remains an area for further research.

7. The study's focus on streaming and non-streaming devices, while valuable, may overlook other categories of IoT devices with unique characteristics. Expanding the research to include a broader range of device types could enhance the model's overall effectiveness and applicability.

Cite this review as

Reviewer 3 ·

Basic reporting

The paper titled "Improved Temporal IoT Device Identification Using Robust Statistical Features” seems a very extensive and detailed study in the IOT domain. This proposed approach utilized the scalable OneVSRest strategy that eases the data labelling process catering the complex and ever-growing IoT System. Again for overtime evaluation this study introduces a weekly dataset segmentation approach, where we used the first week of data for training and the subsequent weeks of data as testing, unlike existing studies that simply train and test on the same timeframe. This study provides an additional dataset for the new benchmark for the study in IoT identification. The paper is clearly written in a good style and includes figures and tables wherever necessary.

Experimental design

This study identifies IoT devices using machine learning techniques and divides the process into
four phases. All the four phases clearly explained the methodology to achieve the objectives of the study.
Some point need to be clarified:
1.The author claimed proposed methodology and proposed feature set have been rigorously tested, this statement needs more clarification with experimental proof .
2Again author claimed to address the complexity associated with many existing IoT device identification approaches. In paper this point need more discussion with valid proof .

Validity of the findings

The authors adequately evaluated their work, and all claims are clearly articulated and supported by empirical experiments.

Additional comments

However, addressing the above comments would improve the quality of the paper. The overall work is good, novel and timely.

Cite this review as

---

## Round 0.2 · accepted · Accept

Congratulations! Based on the positive feedback from our qualified reviewers, your manuscript has been accepted for publication. Please carefully review the next steps.

·

Basic reporting

Based on the previous review, the researchers have removed all my aforementioned considerations, and they have clearly improved the current manuscript. Therefore, I believe that I have no objection to agreeing to publish this version of the manuscript.

Experimental design

ok

Validity of the findings

ok

Additional comments

ok

Reviewer 3 ·

Basic reporting

Not Required

Experimental design

Not required

Validity of the findings

OK

Cite this review as